# Screening Services in a Community Pharmacy in Poznan (Poland) to Increase Early Detection of Hypertension

**DOI:** 10.3390/jcm9082572

**Published:** 2020-08-08

**Authors:** Magdalena Waszyk-Nowaczyk, Weronika Guzenda, Beata Plewka, Michał Michalak, Magdalena Cerbin-Koczorowska, Łukasz Stryczyński, Michał Byliniak, Anna Ratka

**Affiliations:** 1Pharmacy Practice Division, Chair and Department of Pharmaceutical Technology, Poznan University of Medical Sciences, 6 Grunwaldzka Street, 60-780 Poznan, Poland; wguzenda@gmail.com (W.G.); beataszukalska@gmail.com (B.P.); 2Chair and Department of Computer Science and Statistics, Poznan University of Medical Sciences, 7 Rokietnicka Street, 60-806 Poznan, Poland; michal@ump.edu.pl; 3Chair and Department of Medical Education, Poznan University of Medical Sciences, 7 Rokietnicka Street, 60-806 Poznan, Poland; mcerbin@ump.edu.pl; 4Chair and Department of Hypertensiology, Angiology and Internal Medicine, Poznan University of Medical Sciences, ½ Dluga Street, 61-848 Poznan, Poland; l.stryczynski@gmail.com; 5Polish Pharmaceutical Chamber, 16 Długa Street, 00-238 Warsaw, Poland; michal@byliniak.com; 6St. John Fisher College, Wegmans School of Pharmacy, 3690 East Avenue, Rochester, NY 14618, USA; aratka@sjfc.edu

**Keywords:** pharmacist, pharmacy counseling, pharmaceutical care, hypertension, blood pressure measurement, screening services

## Abstract

Background: Community pharmacies in many countries around the world provide healthcare services for patients. Pharmacists trained as medication experts provide a wide range of patient care services related to medication therapy, patient education, disease prevention, and health promotion. Professional training, expertise, and skills qualify pharmacists to engage in health screenings. These screening programs performed by community pharmacists can help to identify risk factors, facilitate early detection of common diseases, and assist physicians with making effective diagnoses. Objectives: In this study, we created and tested a novel model to provide professional monitoring and counseling on blood pressure by community pharmacists. The aims of the study were to identify the prevalence of elevated blood pressure among patients visiting a community pharmacy and describe the demographic characteristics of patients with hypertension (sex, age, education, body weight, and hypertension risk factors). Methods: The research project was conducted in an accredited community pharmacy in Poznan, Poland, from January to April 2019. A total of 118 anonymous patients (30.5% men and 69.5% women) participated in this study. To qualify for this study, participants had to be older than 18 years of age and have no previous diagnosis of hypertension or other cardiovascular disease. Results: Based on the blood pressure screenings, 61.9% of patients were qualified for the standard consultation (SC: normal blood pressure), 21.2% for the intensive consultation (IC: normal blood pressure and hypertension risk factor), 16.9% patients with elevated blood pressure for the high-risk consultation (HRC: referred to a physician), and 3.4% received a diagnosis of hypertension. We qualified 35.6% with a high-pressure value (greater than 140/90 mmHg). Conclusions: The novel model for blood pressure control screening and counseling implemented in a generally accessible community pharmacy may help with early detection of hypertension problems, lead to initiation of effective patient counseling by a community pharmacist, and result in early referral of the patient to a physician.

## 1. Introduction

A globally observed increase in the burden of hypertension is a serious public health problem. Although the percentage of patients treated for hypertension, especially patients achieving the treatment goal, has more than doubled, it is still lower in Poland than the average in western Europe, the USA, and Canada [1]. Therefore, emphasis should be placed on improving early detection, diagnosis, and effective treatment of hypertension, both in basic and specialty care [2].

The incidence of hypertension and its complications is very high in many countries around the world. With the increase in life expectancy and rapid urbanization, even greater prevalence of hypertension is expected in the future. In 2015, 1.13 billion people suffered from hypertension [3]. Reports show that hypertension is controlled in less than one-fifth of patients worldwide who may experience many hypertension health complications, such as myocardial infarction, coronary heart disease, stroke, and atherosclerosis [4]. In Poland, 31.5% of the population has clinically diagnosed hypertension [5]. Reports demonstrate that there is a large group of people diagnosed with hypertension who do not receive any therapy or have ineffective hypertension therapy [6]. More than half of the patients with hypertension are unaware that they need to take antihypertensive medications and do not know how to correctly adhere to the treatment regimen [7]. Therefore, it is critically important to help patients become aware of hypertension and its complications through counseling and regular screenings of blood pressure that can be performed in a community pharmacy.

Counseling provided by pharmacists is an important step toward improvement of disease control and outcomes and, eventually, the overall health and wellbeing of patients, improved quality of life, and better understanding of medicines and diseases [8]. A community pharmacist can play a significant role in the primary prevention of hypertension and other cardiovascular diseases through patient education and counseling, monitoring of medication therapy, management of medication safety, as well as detection and monitoring of specific cardiovascular risk factors. Community pharmacies can be an ideal place to perform health screenings to help with early detection and management of hypertension in the population. Highly qualified pharmacists are capable of engaging in effective healthcare of patients and professional cooperation with patients and physicians to ensure successful therapy outcomes. The involvement of the pharmacist, from early screening and counseling to monitoring of the medication therapy, is the key to achieving positive healthcare outcomes among patients [9].

In Poland, involvement of pharmacists in performing simple diagnostic screenings related to pharmacotherapy is encouraged. Unfortunately, there are no procedures or protocols to guide cooperation between pharmacist and patient during the measurement of blood pressure in a community pharmacy. To address this need, we established our model of blood pressure measurement in a community pharmacy.

We aimed to implement and test the new blood pressure screening model in a community pharmacy setting and assess the prevalence of elevated blood pressure and common risk factors for hypertension among patients visiting the community pharmacy. In addition, we attempted to evaluate the effectiveness of blood pressure screening services provided by community pharmacists.

## 2. Materials and Methods

Research was conducted between January and April 2019 in a community pharmacy in Poznan, Poland. Study participants were randomly selected from patients who visited the community pharmacy as customers and were informed about the possibility of receiving free blood pressure screening. Individuals who agreed to having their blood pressure measured in the pharmacy were enrolled in this study. Participants had to meet two inclusion criteria: over the age of 18 years and no previous diagnosis of hypertension. Each participant was explained the scope of the study and informed that they could resign from further participation at any stage of the research project. A written consent was obtained from each participant. Three pharmacists were familiarized with the study protocol and trained by the investigators on implementation of the patient-counseling model during the blood pressure screenings in the community pharmacy. The new counseling model is presented in Figure A1. The first step of screening protocol was a short semi-structured interview followed by a blood pressure measurement performed with a certified electronic sphygmomanometer with upper-arm cuff (Medel, Check MY 17, Parma, Italy). Each cuff was selected individually for the patient to fit the arm size. The result was an average of 2 measurements with an interval of 1–2 min on the right and left limb. If the difference between the two measurements was greater than 10 mmHg, then a third measurement was taken on the arm that had a higher reading recorded in previous measurement. This reading was included in the average of the obtained measurements. The patient was prepared for examination according to the Polish Society of Hypertension recommendations. Each participant was instructed to rest for at least five minutes in a sitting position with the back supported in a quiet place in the community pharmacy before the measurement. The patient was not included in the study when drinking coffee and/or smoking at least 30 min before the visit. Based on the information obtained in the first step, each patient was assigned to one of three categories: standard consultation (SC: normal blood pressure, without risk factors), intensive consultation (IC: normal blood pressure with risk factors, e.g., family history, elevated blood glucose level and cholesterol), and high-risk consultation (HRC: blood pressure reading ≥ 140/90 mmHg) [10]. Subsequently, participants in each category received specific counseling and intervention in an especially designed, quiet, intimate place in the community pharmacy. SC included basic education of the patient in terms of appropriate lifestyle and prevention of hypertension. For IC, the education was adjusted to the specific risk factor identified in study participants. HRC referred the patient to the physician and provided detailed information on the potential possibility of hypertension with a discussion about risk factors. In each case, the patient received the appropriate educational leaflet. Only patients qualifying for HRC were scheduled for a second follow-up visit in the community pharmacy. If the physician confirmed hypertensive disease, the patient was counseled by a pharmacist and educated about hypertension and self-control of blood pressure.

Participation in this study was anonymous. All collected data were securely stored in the Department of Pharmaceutical Technology, Pharmacy Practice Division at Poznan University of Medical Sciences. The study was conducted in accordance with the Declaration of Helsinki, and the protocol was approved by the Ethics Committee 1146/18.

The Statistica PL 12 (StatSoft) package was used to perform the statistical analysis. The correlations between analyzed nominal data were performed by Chi-square test of independence. All statistical analyses were performed at *p* < 0.05.

## 3. Results

A total number of 118 participants were included in screenings for blood pressure in a community pharmacy (69.5% women and 30.5% men). Table 1 presents characteristics of participants and the post-screening assignments into three consultation categories. During the patient’s visit to the community pharmacy, we measured their blood pressure and followed up with a short interview and counseling with a pharmacist. We assigned patients with normal blood pressure and no additional risk factors (61.9%) to the SC group. Among participants with normal blood pressure, 21.2% had risk factors and we assigned these patients to the IC group. In 16.9% participants, blood pressure was above 140/90 mmHg. These participants were advised and they agreed to be referred to a physician (HRC) (Figure 1). We asked participants to provide their age, height, body weight, and risk factors for developing hypertension. Figure 2 shows that with the increased number of risk factors, the mean blood pressure value increased significantly (*p* = 0.0001). The calculated BMI (body mass index) of patients enabled us to observe a direct proportional relationship between the calculated BMI and the blood pressure values presented in Figure 3 (*p* = 0.0001). A significant relationship between the BMI and age (*p* = 0.0018), and number of risk factors (*p* = 0.001) is presented in Figure 4 and Figure 5, respectively. Among the participants IC group, 44.0% of the patients had one risk factor, whereas 32.0% had two (Figure 6; *p* < 0.0001).

Regarding the HRC patients, 50.0% of participants applied for a follow-up consultation in the pharmacy and 15.0% received recommendations for a more advanced health examination. We diagnosed 20.0% of patients with hypertension (Figure 7). Participants diagnosed with hypertension and increased risk factors for development of hypertension in the future received a higher level of education (*p* = 0.0380; Figure 8) and had higher mean BMI values (Figure 9; *p* = 0.0003). The mean value of systolic and diastolic pressure in people diagnosed with hypertension was the highest among participants referred to a physician and was 164/100 mmHg (Figure 10, *p* = 0.0001 and *p* = 0.0002, respectively). Additional analysis of results by age, sex, and education showed no statistically significant differences.

## 4. Discussion

We focused on community-pharmacy-based screenings of blood pressure in individuals without previously diagnosed hypertension. Among 16.9% participants, we measured blood pressure above 140/90 mmHg. In 2018 during the May Measurement Month, world screening confirmed that 40.5% of screened individuals were not aware of the presence of elevated blood pressure [6]. High blood pressure is often accompanied by various risk factors, e.g., obesity, lipid disorders, or family burden [11,12]. As demonstrated in this study, an increase in the number of risk factors results in higher incidence of hypertension. The incidence of obesity is progressively increasing in Poland, particularly among men [6]. This study showed that the average blood pressure tends to increase with an increase in the BMI, and in participants with BMI ≥ 30.0 kg/m^2^, the pressure was >140/90 mmHg. Positive correlation between BMI values and prevalence of prehypertension and hypertension was previously reported in other studies [13,14,15]. Landi et al. [16] suggested that BMI values correlate with blood pressure values independently from other risk factors. With age, many physical and structural changes occur in the human body, such as decreased function of internal organs, muscle weakness, and changes in body composition (increased body fat) [17]. Similar results were also observed in this study. Higher BMI was associated with an increased number of risk factors.

As a result of blood pressure screenings in a community pharmacy, 16.9% of participants with elevated blood pressure (≥140/90 mmHg) were unaware of it and were referred to a physician. In Poland, almost 40.0% of adults do not know their own blood pressure, and about 30.0% are unaware of having hypertension [10]. Education level showed a correlation with patient reaction to the results of the blood pressure screening. Individuals who had high blood pressure during screening at community pharmacy and were referred to the physician but did not return to the pharmacy with feedback had received a secondary education. We observed that participants with higher education were more aware of the negative effects of hypertension and were more attentive to taking care of their health. They showed the greatest sense of responsibility for their health and confirmed their willingness to reeducate themselves in the pharmacy in the field of diagnosed disease. Literature reports show that higher education correlates with better knowledge and information about hypertension, which in turn results in a healthier lifestyle [18]. On the other hand, the lower socio-economic status, assessed according to the level of education, is associated with a higher incidence of hypertension in young adults [19]. Most people who had an increased risk of developing the disease in the future were obese. Obesity or even excessive weight is inevitably associated with an increased risk of cardiovascular disease. It is also a consequence of comorbidities together with overweight [20]. During the study, we noticed that all the people diagnosed with hypertension by the physician were obese. Of the people referred to the physician, hypertension has since been diagnosed in 20.0% and they have started pharmacotherapy. Their mean blood pressure was the highest among all patients referred to a physician, and their mean BMI was indicative of obesity.

One of the most striking observations to emerge from the data was that out of 78 participants who were younger than 40 years of age and underwent blood pressure screenings, over 14.0% were assigned for HRC. Literature reports confirm that more often younger individuals are diagnosed with hypertension, especially when these individuals have obesity, high levels of uric acid, and hypertriglyceridemia [19,21]. In addition, an increased prevalence of modifiable cardiovascular risk factors in early adulthood (18–30 years old) can result in calcification of atherosclerotic plaques in the coronary arteries, predisposing these young adults to an increased risk of cardiovascular events in the future [22]. This increasing rate of hypertension in young people requires early supervision and prompt treatment to prevent future cardiac events [19].

Early detection of hypertension allows for faster initiation of pharmacotherapy and other treatments. Community pharmacies, due to their easy accessibility to patients and qualified professional staff, are an ideal healthcare setting to perform screenings for the most common diseases. Currently in Polish community pharmacies, health screenings and counseling to improve early disease detection, health promotion, and associated education are unavailable [23,24]. Therefore, implementation of blood pressure screening and counseling services based on professional procedures may lead to the development of standards for controlling blood pressure in a community pharmacy that can be introduced systematically in Poland. Community pharmacists can contribute to the improvement of health outcomes in patients with hypertension and other common chronic disease conditions and eventually, reduction of medical treatment costs that result in significant savings in healthcare costs [25]. Blood pressure screening and counseling services have been provided by community pharmacies for a long time in many countries and should be implemented and made available to everybody in Polish community pharmacies.

## 5. Limitation of the Study

This screening program included a relatively small sample number of subjects older than 50 years of age, which is quite limited. One of the reasons may be adequate high prevalence of already diagnosed hypertension in this population. It should be emphasized that patients under the age of 40 years, as a group less exposed to chronic diseases, are at greater risk of suffering from undetected hypertension. Younger patients also ask for medical consultations less often. In the obtained results, over 14.0% of patients were qualified for the HRC, which seems to confirm the legitimacy of implementing screening services in places more accessible to patients, such as community pharmacies. Finally, data were collected in a single community pharmacy, so more detailed studies are needed in the future.

## Figures and Tables

**Figure 1 jcm-09-02572-f001:**
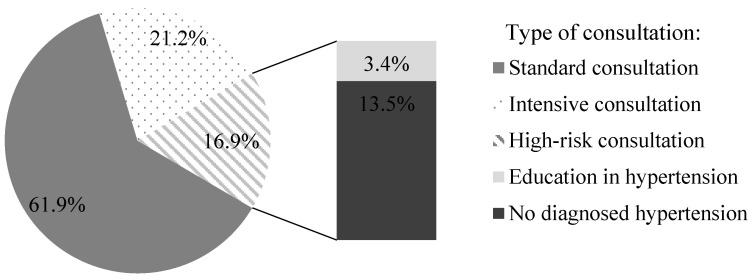
Distribution of study participants into three consultation groups based on the results from the blood pressure screening in a community pharmacy (*n* = 118).

**Figure 2 jcm-09-02572-f002:**
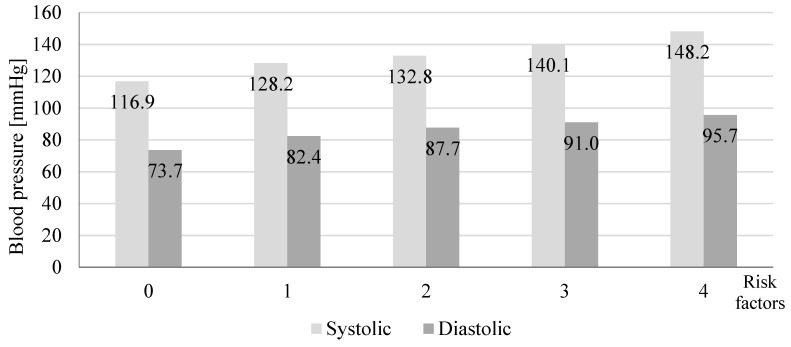
Effect of the number of risk factors on the mean value of systolic and diastolic blood pressure in the study participants screened in a community pharmacy (*n* = 118, *p* = 0.0001).

**Figure 3 jcm-09-02572-f003:**
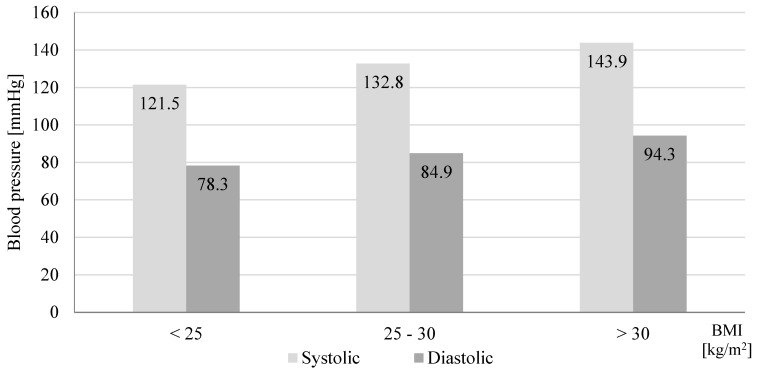
Effect of body mass index (BMI) on mean value of systolic and diastolic blood pressure in participants screened in a community pharmacy (*n* = 110, *p* = 0.0001).

**Figure 4 jcm-09-02572-f004:**
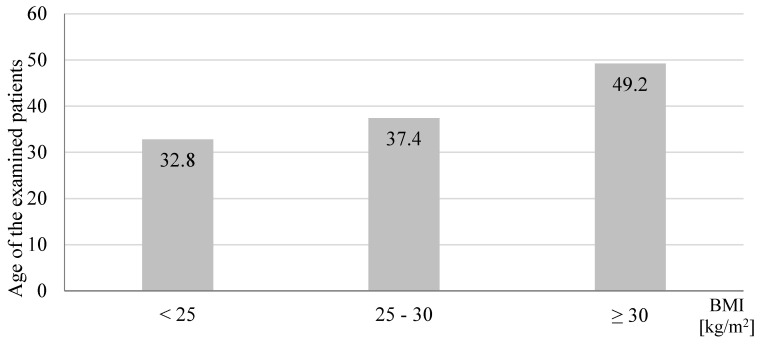
Effect of age on body mass index (BMI) in study participants screened for blood pressure in a community pharmacy (*n* = 110, *p* = 0.0018).

**Figure 5 jcm-09-02572-f005:**
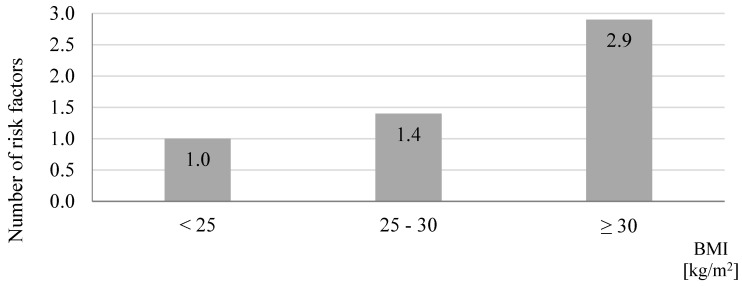
Average number of risk factors in relation to the body mass index (BMI) in study participants screened in a community pharmacy (*n* = 110, *p* = 0.0001).

**Figure 6 jcm-09-02572-f006:**
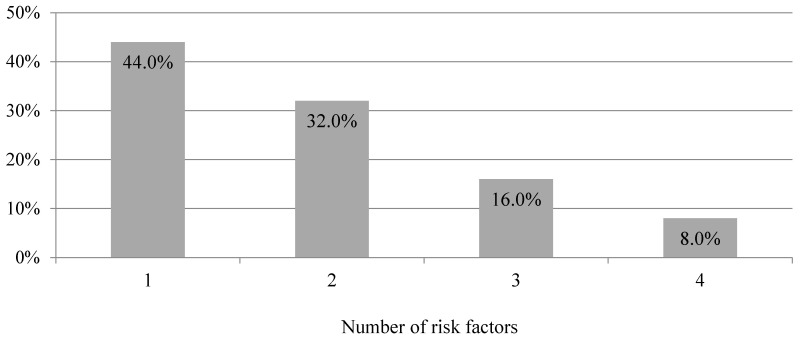
Number of risk factors among study participants recommended to intensive consultation after blood pressure screening in a community pharmacy (*n* = 25, *p* < 0.0001).

**Figure 7 jcm-09-02572-f007:**
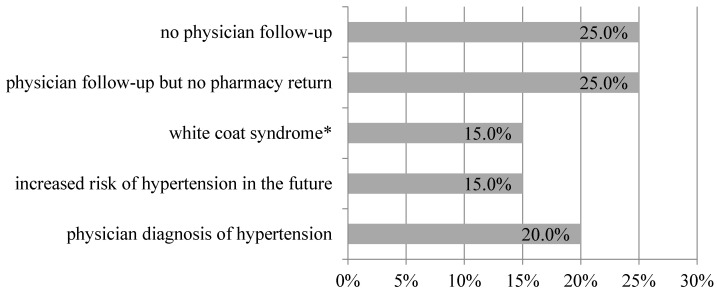
Characteristics of individuals classified into the high-risk consultation group and referred to a physician after blood pressure screening in a community pharmacy (n = 20). * patient with elevated blood pressure upon physician visit, but with normal results at home.

**Figure 8 jcm-09-02572-f008:**
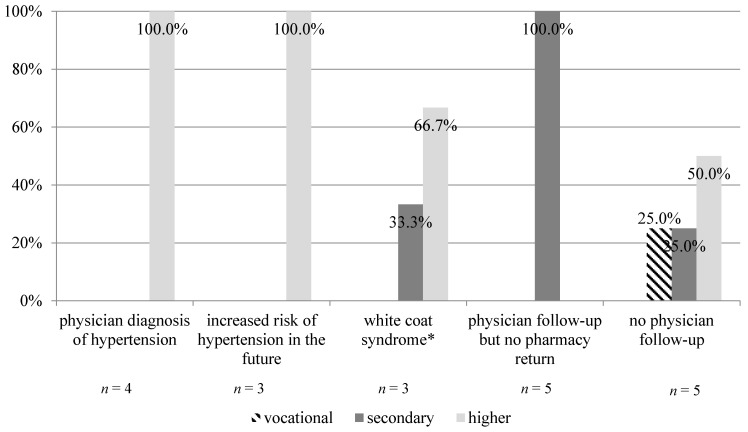
Effect of education level on outcomes from blood pressure screening among study participants who received a high-risk consultation (n = 20, *p* = 0.0380). * patient with elevated blood pressure upon physician visit, but with normal results at home.

**Figure 9 jcm-09-02572-f009:**
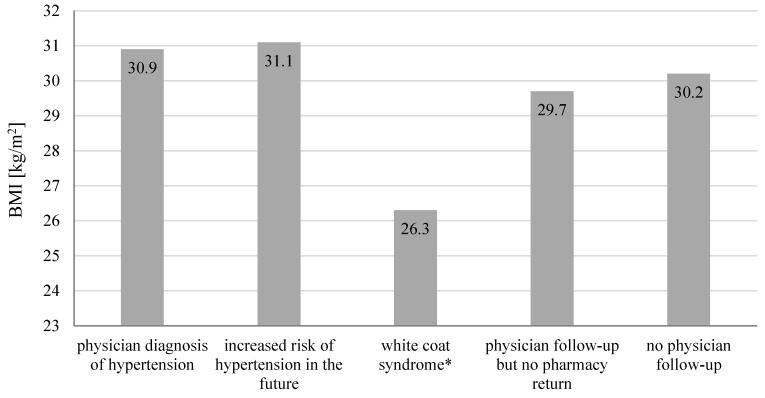
Effect of the average Body Mass Index (BMI) on the outcomes from study participants who received a high-risk consultation (n = 20, *p* = 0.0003). * patient with elevated blood pressure upon physician visit, but with normal results at home.

**Figure 10 jcm-09-02572-f010:**
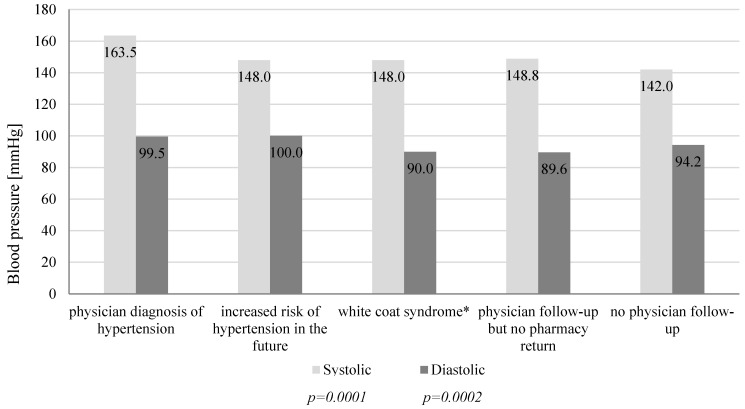
Effect of mean value of systolic and diastolic blood pressure on the outcomes among study participants who received a high-risk consultation (n = 20). *patient with elevated blood pressure upon physician visit, but with normal results at home.

**Table 1 jcm-09-02572-t001:** Characteristics of participants involved in study on blood pressure screening in a community pharmacy (*n* = 118).

Examined Patients *n* (%)									
Type of Consultation	SC Standard Consultation		IC Intensive Consultation		HRC High-Risk Consultation		Total		
Sex	Male	Female	Male	Female	Male	Female	Male	Female	Total
**Total Respondents**	16 (13.6)	57 (48.3)	11 (9.3)	14 (11.9)	9 (7.6)	11 (9.3)	36 (30.5)	82 (69.5)	118 (100)
Age [years] *n* = 118									
19–29	4 (25.0)	24 (42.1)	1 (9.1)	4 (28.7)	1 (11.1)	3 (27.3)	6 (16.7)	31 (37.8)	37 (31.4)
30–39	6 (37.5)	21 (36.8)	6 (54.5)	1 (7.1)	4 (44.5)	3 (27.3)	16 (44.5)	25 (30.5)	41 (34.7)
40–49	3 (18.7)	12 (21.1)	3 (27.3)	5 (35.7)	2 (22.2)	2 (18.1)	8 (22.2)	19 (23.1)	27 (22.9)
50–59	2 (12.5)	0 (0.0)	1 (9.1)	3 (21.4)	0 (0.0)	0 (0.0)	3 (8.3)	3 (3.7)	6 (5.1)
≥60	1 (6.3)	0 (0.0)	0 (0.0)	1 (7.1)	2 (22.2)	3 (27.3)	3 (8.3)	4 (4.9)	7 (5.9)
**Education** ***n* = 111**									
High school	1 (6.7)	1 (1.9)	0 (0.0)	0 (0.0)	0 (0.0)	0 (0.0)	1 (3.0)	1 (1.3)	2 (1.8)
Vocational	0 (0.0)	0 (0.0)	1 (9.1)	2 (14.3)	1 (12.5)	0 (0.0)	2 (5.9)	2 (2.6)	4 (3.6)
Secondary	5 (33.3)	11 (21.2)	1 (9.1)	7 (50.0)	2 (25.0)	5 (45.5)	8 (23.5)	23 (29.9)	31 (27.9)
Higher	9 (60.0)	40 (76.9)	9 (81.8)	5 (35.7)	5 (62.5)	6 (54.5)	23 (67.6)	51 (66.2)	74 (66.7)

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
