# Peer review of "Screening Services in a Community Pharmacy in Poznan (Poland) to Increase Early Detection of Hypertension"

_jcm, 2020, doi:10.3390/jcm9082572_

Round 1

Reviewer 1 Report

Brief Summary

This study achieves the aim of evaluating the prevalence and risk factors for hypertension noted in a population participating in novel community pharmacy blood pressure screening in Poland. The authors describe the contribution community screening can make to early detection of hypertension and risk factors with a standardized approach that can be replicated across community pharmacies. By providing this screening the authors do report identification of 40% of their population as having either risk factors for hypertension or elevated blood pressure at screening, thus prompting physician referral. I also think the authors highlight potential value in pharmacist follow-up visits with this investigation. I think this provides justification to expand this process/protocol and would be interested to see future data looking at an algorithm to assist pharmacists in identifying patients to screen, patient-satisfaction with screening, and incorporating physician response and satisfaction with community pharmacy screening and patient referral process.

Broad Comments

Strengths:

  • Describes a proof-of-concept of pharmacist screening in a community pharmacy setting to identify risk factors and possible presence of undiagnosed hypertension.
  • Provides an opportunity for the community pharmacist to both educate their patients and provide an accessible service to identify early-stage hypertension or at-risk patients.
  • Provides avenue for further study identifying both patient satisfaction with screening as well as potentially a survey/future study on physician attitudes for patients referred due to pharmacist screening.

Weaknesses:

  • The authors indicate patients received specific counseling based on the patient risk category (SC, IC, HRC). The authors cite the 2019 Guidelines for the Management of Hypertension from the Polish Society of Hypertension. As a reader I think it would be helpful to have some information regarding the specific details covered in counseling, and how SC differed from IC, briefly in the text or as a table/appendix.
  • The authors report in Figures 7-10 characteristics for the HRC group. I like seeing the outcomes with this high-risk group and have some specific comments below

Specific Comments

  • Page 2, Line 78, “Randomly selected patients” – did the authors select patients at random when that patient came to the community the pharmacy OR did the authors identify random patients from within the community pharmacy system and contact them for the study?
  • Page 2, Line 91 -> Categories of consultation, the authors report details of SC and IC, but not the details of what HRC qualifications are. The Appendix 1 indicates HRC was provided to patients with a BP reading ≥140/90 mm Hg, I would add this to the text as well.
  • Page 3, Line 97 – Recommend wording adjustment to “Only the patients qualifying for HRC were scheduled for a second, follow-up visit in the community pharmacy.” (or something similar to more clearly state only HRC patients were offered a second visit, will help make Figures 7-10 more clear.
  • Page 3, Line 122, would introduce this paragraph with “Of the HRC patients, 50% of participants applied…” to make it clear we are shifting to the HRC group only for follow-up visit data.
  • Figures 7-10: These follow-up categories are interesting/useful to illustrate pharmacist impact, a few comments:
    • Reword people who didn’t got to the doctor to “no physician appointment after initial HRC” (or something similar, for clarity and consistent use of the term physician)
    • ‘people with the white coat syndrome’ – does this mean patients without elevated blood pressure upon physician visit or without elevated blood pressure on repeat pharmacy visit?
      • If the term white coat syndrome is to be used you may wish to clarify the definition you are using in the text?
    • Is the category ‘people with diagnosed hypertension’ mean those who saw a physician and now have a new/official diagnosis? If so consider “people with physician diagnosis of hypertension”
    • May want to remove ‘People who’ and just state “no follow-up with physician’, ‘physician follow-up but no pharmacy follow-up’, ‘diagnosed with hypertension’ (but that is my preference to use shorter terms more so than a necessary change)

Reviewer 2 Report

This is a small report in one pharmacy of a screening program for high blood pressure. It is admirable that the authors initiated this program and created this report of their experience. On the downside, the results that they found some hypertensive subjects is not surprising, and the incidence of hypertension they report is actually lower than what is often observed in large Western populations.

Please state more completely how the screening blood pressure measurements were performed. In particular, were the subjects allowed to rest for at least 5 minutes? Was only one reading obtained?

The number of subjects is small. How can you be sure this is representative?

Related to above, it is disappointing that the number of subjects older than 50 and 60 is so small, since the incidence of hypertension increases with age. Does this reflect measurement bias that older subjects don’t visit the pharmacy? This should be discussed.

Author Response

This manuscript is a resubmission of an earlier submission. The following is a list of the peer review reports and author responses from that submission.

Round 1

Reviewer 1 Report

This study analyzed the blood pressure levels and characteristics of 118 adults who were screened for hypertension in a community pharmacy. The authors concluded that this model for blood pressure screening may help with early detection of hypertension and referral to physicians.

COMMENTS

  1. Methods: ‘Randomly selected patients were informed about the possibility to receive free blood pressure screening...’. Please provide detailed description of the ‘random selection procedure’.
  2. Abstract: if 61.9% had normal blood pressure and 21.2% normal blood pressure and hypertension risk factor, then a a total of 83.1% with normal blood pressure. How is that 35.6% had high blood pressure? In Line 108-109 it is stated that in 16.9% participants blood pressure was >140/90 mmHg.
  3. How many blood pressure readings were obtained? Please provide information of the setting, conditions, body position, number of blood pressure measurements, and averaging method.
  4. Device used for measuring blood pressure. Is this auscultatory or electronic? Upper-arm cuff or wrist-cuff? How many cuffs were available and how they were selected to fit the arm size of each individual?
  5. Please provide reference for published validation study documenting the accuracy of. This device is not included in the lists recommended by the www.stridebp.org.
  6. This screening program included a relatively small sample of mostly young adults and 70% women. Thus, the prevalence of hypertension is not expected to represent the general population of adults and was certainly underestimated
  7. Line 91: Please replace ‘incorrect’ with ‘elevated’.